# CLUMM: Contrastive Learning for Unobtrusive Motion Monitoring

**DOI:** 10.3390/s25041048

**Published:** 2025-02-10

**Authors:** Pius Gyamenah, Hari Iyer, Heejin Jeong, Shenghan Guo

**Affiliations:** 1The School of Manufacturing Systems and Networks, Ira A. Fulton Schools of Engineering, Arizona State University, Mesa, AZ 85212, USA; pgyamena@asu.edu; 2The Polytechnic School, Ira A. Fulton Schools of Engineering, Arizona State University, Mesa, AZ 85212, USA; hniyer@asu.edu (H.I.); heejin.jeong@asu.edu (H.J.)

**Keywords:** self-supervised learning, unobtrusive human sensing, contrastive learning, in situ monitoring, motion recognition

## Abstract

Traditional approaches for human monitoring and motion recognition often rely on wearable sensors, which, while effective, are obtrusive and cause significant discomfort to workers. More recent approaches have employed unobtrusive, real-time sensing using cameras mounted in the manufacturing environment. While these methods generate large volumes of rich data, they require extensive labeling and analysis for machine learning applications. Additionally, these cameras frequently capture irrelevant environmental information, which can hinder the performance of deep learning algorithms. To address these limitations, this paper introduces a novel framework that leverages a contrastive learning approach to learn rich representations from raw images without the need for manual labeling. This framework mitigates the effect of environmental complexity by focusing on critical joint coordinates relevant to manufacturing tasks. This approach ensures that the model learns directly from human-specific data, effectively reducing the impact of the surrounding environment. A custom dataset of human subjects simulating various tasks in a workplace setting is used for training and evaluation. By fine-tuning the learned model for a downstream motion classification task, we achieve up to 90% accuracy, demonstrating the effectiveness of our proposed solution in real-time human motion monitoring.

## 1. Introduction

Human motion recognition is a task that identifies human motion from sensor data, with significant implications across domains such as healthcare [1,2], sports [3], and manufacturing [4], where it is essential to understand human behavior from motion patterns. For instance, in the manufacturing domain, it can provide valuable insights for understanding worker behavior, identifying potential safety risks, assessing workers, detecting ergonomic issues, and identifying areas where further training may be needed. Despite this potential, monitoring and analyzing human motion requires significant effort in sensing and analysis [5], especially in complex environments with many moving parts.

Most existing sensor-based human motion recognition efforts have utilized wearable sensors worn directly on body parts [6]. While wearable sensor technologies have advanced significantly, they are intrusive and may need to be adjusted to different user heights and sizes. Additionally, multiple wearable sensors must be used simultaneously in many cases, which can be cumbersome, cause discomfort, and reduce worker productivity [6,7,8].

Advancements in camera technology and image processing have paved the way for unobtrusive in situ monitoring. Comparative studies from the literature show that camera-based approaches offer a less invasive and more comprehensive solution [9,10]. These camera-based approaches can capture detailed motion data without interfering with human activities, thus facilitating real-time monitoring and analysis. However, analyzing these large streams of real-time data comes with significant data-level challenges. Recent advances in deep learning (DL) have paved the way for more accurate human activity recognition from sensor data [8,9,10,11,12]. Deep learning approaches such as neural networks (NNs) can extract features directly from input data, thus providing end-to-end learning on raw sensor data without extensive preprocessing. Nevertheless, their performance heavily depends on large volumes of labeled data [13]. Unobtrusive camera-based sensors provide large data streams, but assigning activity labels individually to these large streams of sensor data is costly and labor-intensive and requires significant domain expertise, posing significant barriers to the efficiency and scalability of existing solutions. Thus, there is a need for label-efficient approaches to address this data-level bottleneck. Additionally, the dynamic environments in which human motion is monitored challenge the effectiveness of learning algorithms due to various factors such as occlusions, noise, varying lighting conditions, and irrelevant objects in the scene [7]. These factors can lead to the Clever Hans phenomenon [14], where a model performs well but relies on irrelevant data (spurious correlations) instead of learning the task of interest. This phenomenon can manifest in neural-network-based worker motion monitoring from image data. An algorithm may associate irrelevant features, such as background machinery and lighting conditions, with the motion task instead of focusing on human movements. This problem affects the generalizability of machine learning solutions to complex environments.

Recently, label-efficient approaches, such as self-supervised learning (SSL), which learn representations directly from unlabeled data, have been proposed to overcome the limitations posed by the lack of labels [15,16]. SSL approaches such as contrastive learning (CL), which leverage instance discrimination to bring similar instances closer in the representation space while pushing dissimilar instances apart, have demonstrated superior performance across various modalities with robust generalizability compared to supervised approaches [16]. Additionally, they are less susceptible to spurious correlations and adversarial examples. Due to their label efficiency, robustness to variations, and generalizability, these approaches hold immense potential for unobtrusive human motion recognition tasks. However, to the best of our knowledge, they have not been explored enough for unobtrusive motion recognition. Existing self-supervised approaches to human motion recognition have focused on wearable sensors [17,18].

To enable label-efficient human motion recognition from unobtrusive sensing data, we introduce **Contrastive Learning for Unobtrusive Motion Monitoring (CLUMM)**, a contrastive SSL-based framework for unobtrusive human recognition, leveraging the robust feature extraction and generalization capabilities of SSL to learn representations directly from unlabeled camera data. As shown in Figure 1, we extract skeletal coordinates from image frames of human videos and use them as features to learn representations from them without manual data labeling. We show the effectiveness of the proposed approach by fine-tuning it on a small dataset of labeled human motion data. CLUMM addresses the data-level challenges in existing unobtrusive motion recognition methods as follows:

**Joint tracking by Computer Vision (CV).** We remove the effect of the complex operational environment by directly extracting the coordinates of specific joints from the human body using CV techniques, specifically MediaPipe pose (MPP) [19]. MPP is an open-source framework developed by Google for estimating high-fidelity 2D and 3D coordinates of body joints. It uses BlazePose [20], a lightweight pose estimation network, to detect and track 33 3D body landmarks from videos or images. From the body landmarks identified by MPP, we select key landmarks based on the inputs of ergonomic experts to formulate the initial features significant to various motion types. This method of extracting joint information also preserves privacy by learning from joint coordinates instead of raw image data. Additionally, the MPP is scale- and size-invariant [19], which enables it to handle variations in human sizes and height.**SimCLR feature embedding.** We address the data bottleneck using a contrastive SSL approach to directly learn representations from camera data without requiring extensive manual labeling. We specifically use SimCLR [21], an SSL method that learns features by maximizing agreement between different augmented views of the same sample using a contrastive loss. We use SimCLR to learn embeddings and identify meaningful patterns and similarities within the extracted joint data depicting various motion categories. The learned representations are further leveraged in a downstream task to identify specific motion types.**Classification for motion recognition and anomaly detection.** Lastly, we leverage the learned representations from the CL training for a downstream classification task involving different motion categories. We train a simple logistic regression model on top of our learned representations to identify different motion categories in a few-shot learning [22] setting. This demonstrates the robustness and generalizability of our learned representations to downstream tasks. Additionally, we perform outlier analysis by evaluating the ability of our framework to identify out-of-distribution data. We introduce different amounts of outliers with varying deviations from the classes of interest and measure the ability of our framework to identify these outliers and the effect of outliers on the discriminative ability of our framework.

The proposed CLUMM contributes to the methodology and application of unobtrusive human motion monitoring. Methodologically, CLUMM will contribute a label-efficient machine learning approach to recognize motion types from unobtrusive human sensor data. CLUMM’s ability to remove the effects of the complex environment will make it applicable to motion monitoring in such environments. Furthermore, CLUMM will contribute to anomaly detection and outlier analysis across various motion analysis tasks due to its robustness to outliers, as demonstrated in our case study.

Practically, CLUMM is highly useful for worker motion analysis. By integrating MPP as its joint feature extraction component, which is then connected with SimCLR and motion classification, CLUMM achieves improved feature extraction, adaptability to different downstream tasks, and reduced manual labeling efforts. Additionally, CLUMM is robust to outliers and can capture other motion types inherent in the training data. Our approach to worker motion analysis presents a step toward effective and efficient human motion analysis. In a case study, we show the effectiveness of our solution by fine-tuning domain-specific data involving various task categories in a controlled laboratory environment. Using a few labeled examples, CLUMM outperforms transfer learning performance on a baseline ResNet model, demonstrating the superiority of the learned representations as a feature extractor for other tasks in a similar domain.

The rest of this paper is organized as follows. Section 2 provides an overview of state-of-the-art literature. Section 3 elaborates on the technical details of our proposed methodology. Section 4 presents a case study using domain-specific data from manufacturing tasks in a laboratory environment and provides a discussion of our results and highlights future directions. Finally, Section 5 concludes the work.

## 2. Literature Review

Human motion recognition is a task that identifies human motion automatically from sensor data [12,23,24]. This rapidly growing field is significant to domains such as manufacturing, where it is necessary to understand human behavior, assess workers, provide additional training, and suggest ergonomic improvements [5]. Human motion recognition involves analyzing collected sensor data to identify and classify various motion categories [25,26,27]. Machine learning and deep learning approaches have been utilized to analyze sensor data using supervised and unsupervised methods. Supervised approaches, however, require large volumes of annotated data, which is both time- and labor-intensive. This limitation has led to the exploration of label-efficient approaches [28], such as self-supervised learning [15,16]. This section reviews existing work on unobtrusive human motion sensing and self-supervised learning, highlighting the methodological gaps and providing justifications for our proposed approach.

### 2.1. Unobtrusive Human Motion Monitoring

Unobtrusive sensing makes sensors as invisible as possible by blending sensors into the natural environment. This invisibility enables users to perform their activities unobtrusively and non-invasively [6]. Radio Frequency Identification (RFID) [4,29], thermal cameras [8,11,30], millimeter-wave (mmWave) radar [29], LiDAR [20], and combinations of different non-wearable sensors with multimodal data [31] have been used in the literature for recognizing different motion categories across various domains. Beyond comfort, unobtrusive sensing using nonwearable sensors blended in the environment provides advantages such as broader coverage (cameras), consistency, and reduced signal noise [9,32,33,34,35]. Additionally, no user training on using the sensor is needed.

In recent years, motion sensor analysis has seen substantial improvements, reflecting various methodological approaches that leverage traditional rule-based approaches, classical machine learning, deep learning, and a mixture of preprocessing and deep learning.

Early approaches to human motion analysis primarily utilized rule-based mathematical and interpolation approaches based on domain expertise to analyze and classify various motion categories [33,34]. These methods involved the application of numerical algorithms to process and interpret sensor data. For example, Newaz et al. [33] used interpolation and mathematical measures to analyze data obtained from an array of low-resolution thermal sensors. Results from their study demonstrated the effectiveness of these rule-based approaches, showing superior performance to some machine learning (ML) approaches. In [34], the authors leveraged skeleton coordinates from a Microsoft Kinect sensor for classifying activities such as standing, sitting, lying, and falling. They performed handcrafted feature extraction using depth, height, velocity, acceleration, and angle between joints to classify various action categories. They set a threshold for determining various classes of activities and manually compute motion categories by comparing extracted features against the given thresholds for each action type. While these rule-based approaches have been proven to work with more straightforward datasets and tasks, they depend solely on predefined rules, which are prone to errors and may fail to capture complex patterns and relationships in complex sensor data where explicit rules may not be easily inferred. Additionally, they depend heavily on domain expertise, which may not always be readily available. Nevertheless, these approaches can be starting points for more efficient ML/DL approaches. For instance, the handcrafted features from [34] can serve as features for DL algorithms.

Several works in motion recognition have utilized classical machine learning approaches such as hidden Markov models, support vector machines (SVM), k nearest neighbors (KNNs), and ensemble methods such as random forests (RF) [27,28,29,36,37,38] to classify and predict motion patterns from sensor data. These approaches use manual feature extraction to extract relevant motion features and use them as inputs to train predictive models. In [37], the authors used a maximum entropy Markov model (MEMM) for activity recognition using a modified Viterbi algorithm to model the most probable activity sequences after preprocessing and feature extraction with optical flow and stepwise linear discriminant analysis. They modeled the activity states as states of the MEMM and identified the next activity sequence based on video input. Liu et al. [4] use a similar Markov approach to predict the next probable motion sequence in a human–robot collaborative assembly task. KNNs, RFs, and SVMs are used in [28] on data from an array of thermopile sensors to classify 15 human motion categories. Their results demonstrate the effectiveness of classical ML approaches.

Despite their efficacy, these classical ML approaches often need manually extracted handcrafted features from sensor data and activity labels as inputs to classifiers. They require extensive domain knowledge for feature engineering and may struggle with high-dimensional data and complex motion patterns [12]. Unlike classical machine learning approaches, deep learning approaches such as neural networks can extract features directly from input data, thus providing end-to-end learning on raw sensor data without extensive preprocessing. Additionally, deep learning approaches are scalable to large volumes of sensor data and adaptable to different scenarios, making them suitable for the motion recognition task [39]. Rezaei et al. [8] show the effectiveness of deep learning approaches by comparing the results of applying ML and DL algorithms to data from an array of low-resolution infrared cameras. The infrared cameras were mounted on an experimental room’s side wall and ceiling. The data in stereo images were used as ground truths for ML and DL algorithms. Their results showed the superiority of deep learning methods over traditional ML approaches. Additionally, they show the advantage of sensor fusion in this domain by comparing the results from a single sensor and a fusion of multiple infrared sensors at different locations. Multiple comparative studies [29,40] have also shown the superiority of DL approaches for motion recognition. Convolutional neural networks (CNNs) and LSTMs have particularly demonstrated remarkable performance; they have been used extensively in recent work [7,8,9,10,11,30,41,42,43] as well as in prior work, with impressive recognition performance showing their ability to capture spatiotemporal dependencies in motion data.

Recent advancements have explored hybrid approaches that combine preprocessing techniques with deep learning models. These approaches employ traditional preprocessing steps such as denoising, voxelization, and dimensionality reduction to enhance sensor data quality before feeding them into deep learning models. For instance, Singh et al. [29] used a mixed approach using a preprocessing approach and ML and DL approaches. They process the raw data from an mmWave radar using a sliding window to gather point clouds and convert them into voxels. These voxels are used as inputs for different classifiers. They evaluate the performance of various classifiers such as SVMs, Multi-layer Perceptron (MLP), Bi-Directional LSTM, and Time-Distributed CNN + Bi-directional LSTM on the processed inputs and find that the Time-Distributed CNN + LSTM performs the best. Similarly, Yu et al. [9] used DBSCAN to remove noise from point clouds. They conducted voxelization and augmented the voxelated input before feeding it into a dual-view CNN for end-to-end learning. These integrations demonstrate the effectiveness of harnessing the strengths of preprocessing methods and advanced deep-learning algorithms to achieve motion recognition performance.

These hybrid approaches have been particularly transformative in camera-based motion analysis. In this approach, information about the positions and movement of body parts extracted from pose estimation models is input into deep learning models. Utilizing this joint information allows for a more coherent representation of motion dynamics as it captures the intricate relationships between body parts and their movements over time [44]. In [45], the authors classify activities using the Euclidean distance of 3D joint coordinates in consecutive frames as input to a CNN for activity classification.

A comprehensive comparative study by Açıs et al. [40] sought to evaluate training effectiveness using raw RGB data from a Kinect sensor versus utilizing pose coordinates. The study assessed the performance of an LSTM feature extractor and compared it with feature extraction using a CNN on joint coordinates. Additionally, the study compared the effectiveness of training a CNN from scratch and using three transfer learners (Densenet201 [46], Xception [47], and Resnet50 [48]) on raw RGB images. The comparative evaluation results indicated that utilizing joint coordinates and an LSTM feature extractor yielded the best accuracy. Specifically, joint data with a CNN from scratch resulted in 72% accuracy. In comparison, joint data with LSTM achieved 98% accuracy, and the best transfer learner on raw RGB images produced an accuracy of 60%. Thus, the study concluded that utilizing joint information for training yielded significantly better results than training on raw RGB images. Inspired by the results from [40], this study utilizes pose information as input for feature extraction.

Despite the advantages of deep learning-based approaches to the motion recognition task, their performance heavily depends on large volumes of labeled data [13], which is time-consuming and challenging to annotate. Self-supervised approaches to motion recognition seek to address this data-level challenge by learning representations directly from unlabeled data without relying on class labels. This is crucial as labeling motion data is labor-intensive and time-consuming, especially for diverse and complex motions. Additionally, human–subject data are expensive to gather in practice [1], requiring remunerations and IRB approvals in some cases. These limit the data available for training these DL algorithms for new sensor data. Thus, there is a need for label-efficient and adaptive approaches to learning robust representations that can be fine-tuned on small sensor data as transfer learners.

Self-supervised approaches to motion recognition employ various strategies to learn representations from motion data without manual annotations [49,50,51,52,53]. Contrastive learning is widely used for its ability to create robust backbone networks by contrasting spatiotemporally augmented video clips [51,52,53]. For example, Lin et al. [50] adopts a multi-task perspective, leveraging pretext tasks such as jigsaw puzzles and motion recognition to enhance diversity in skeleton-based representations, though these methods can be computationally expensive. Simpler approaches [52] avoid pretext tasks entirely, focusing on extracting principal information about actions in videos by contrasting augmented video clips with the original, refining latent representations without relying on negative examples.

Yang et al. [51] formulates the motion recognition task as a “self-supervised motion attention prediction problem” using skeleton coordinates. They eliminate the need for augmentations and instead utilize attention maps obtained from motion videos to estimate joint importance in spatial and temporal dimensions. Their method maximizes agreement between the attention maps and pseudo labels generated from prior knowledge, thereby learning representations for downstream motion recognition tasks. Similarly, Seo et al. [53] introduces a self-supervised sampler to select relevant frames from motion recognition videos. This method leverages contrastive learning to differentiate irrelevant frames from motion sequences, providing a foundation for other motion recognition approaches. While complex approaches like multi-task and attention-guided methods offer diverse and detailed representations, they often involve trade-offs between model complexity and adaptability to real-world constraints.

We address this by leveraging SimCLR, a contrastive self-supervised learning framework, to learn representations directly from limited unlabeled data with a simple and straightforward approach. The framework can learn robust representations that can be used for downstream tasks in human motion-related tasks on sensor data.

### 2.2. Self-Supervised Learning

As discussed in Section 2.1, existing approaches to human motion recognition face significant challenges due to their reliance on large volumes of labeled data, which is expensive to collect and requires a substantial manual labeling effort and domain expertise. Self-supervised learning (SSL) emerges as a strong alternative to address this data bottleneck.

Self-supervised Learning (SSL) is a machine learning paradigm that learns representations directly from unlabeled data without relying on corresponding class labels [15]. SSL addresses the data bottleneck of supervised learning approaches, thereby alleviating the reliance on large, labeled datasets. SSL has recently received much attention and contributed significantly to advances in natural language processing (NLP) and CV [16]. Recent work has shown the effectiveness of SSL approaches across different data modalities such as text, audio, video, and time series [54,55,56]. SSL leverages unlabeled inputs to define a pretext task, which it uses to learn representations [57]. These representations capture the internal relationship between inputs, which can be fine-tuned for downstream tasks [57]. In contrast to supervised learning approaches, where representations are task-specific, the representations learned by SSL are task-agnostic [15], making the learned representations adaptable to other tasks. Additionally, SSL boasts of better generalizability and robustness to spurious correlations and adversarial attacks than supervised learning [15,16].

Existing approaches to SSL employ different strategies to accomplish the pretext task, which involves learning supervisory signals from unlabeled data. Context-based approaches [15] utilize the intrinsic contextual relationships within the inputs, such as color and rotations, as supervisory signals. These approaches train models to understand the positions and orientations of objects within a scene [16]. On the other hand, generative approaches [15] focus on reconstructing input data or generating new ones. These approaches involve masked image modeling (MiM) [58], where large portions of images are masked for the network to repaint. This approach focuses on local views and pays particular attention to internal information.

CL approaches build on instance discrimination, bringing similar instances closer in the representation space while pushing dissimilar instances apart. CL approaches offer better discriminative power and more robust feature representation learning than other approaches, as they are trained to differentiate between different data instances [21]. They do not require complex reconstruction tasks, making them simple to train and implement. They are also scalable to large data, unlike generative models. A range of methods have been used in the literature to contrast data samples.

Negative example-based CL treats views originating from the same sample as positive pairs while treating views from different instances as negative pairs. These methods maximize the proximity between positive pairs and the separation between negative pairs [15]. This method is used in MoCo and SimCLR [21] for 2D image CL.

Self-distillation-based methods like Bootstrap Your Own Latent (BYOL), Simple Siamese Networks (SimSIAM), and DINO eliminate the need for negative pairs. They feed two different views of an input sample to two similar neural network encoders with the same architecture but different weights. These networks are then mapped to each other using a predictor. These methods maximize the similarity between positive pairs while employing diverse strategies to prevent mode collapse [15,16].

Feature decorrelation-based CL methods like Barlow Twins are designed to learn decorrelated features. These approaches generate distorted views of the same image using a distribution of data augmentations and employ strategies to encourage similarities within the embeddings of distorted views while minimizing redundancy between the features. Akin to self-distillation approaches, methods are also used to prevent collapse [15].

Chen et al. [21] introduced SimCLR, a straightforward negative example-based contrastive learning framework for learning feature representations. SimCLR simplifies contrastive learning using a straightforward approach that reduces the need for high memory. It achieves this by contrasting transformed views of images in the same batch and allows for flexible batch sizes [1]. SimCLR is architecturally simple. It utilizes well-known augmentations, simple projection heads, and encoders, making it scalable and adaptable to other architecture and data modalities. It is also model agnostic, allowing for easy application to sensor data by adjusting the encoder to the specific sensor data [12]. Beyond its simplicity and flexibility, SimCLR achieves state-of-the-art performance on benchmark tasks. We use SimCLR as a representation learner in our proposed model owing to its simplicity, robustness, and scalability to different data modalities.

### 2.3. Section Summary

This section has explored various unobtrusive human motion analysis approaches, including mathematical and rule-based approaches, classical machine learning, deep learning, and hybrid approaches. Although each method has contributed unique strengths and driven advancements toward human motion analysis, they all share a common challenge: the need for large volumes of labeled data, which is costly and requires skilled human annotation with sufficient domain expertise. This data-level challenge is pertinent in human motion analysis, where multiple sensors can easily obtain large data streams. However, assigning activity labels individually to these large streams of sensor data is costly and labor-intensive, posing significant barriers to the efficiency and scalability of existing solutions. We address this gap in this study by harnessing the ability of self-supervised learning to learn supervisory signals directly from unlabeled sensor data. We introduce a hybrid approach combining data preprocessing with SimCLR, a straightforward contrastive learning framework to learn robust representations from video streams. Our learned representations can be used to fine-tune models for downstream tasks. By addressing this data-level gap, we alleviate the financial and logistical burden associated with data labeling and enhance the scalability and robustness of unobtrusive human motion recognition.

## 3. Method Development

In this section, we outline the methodological details of CLUMM. As illustrated in Figure 1, the proposed CLUMM framework first extracts human pose landmarks from image frames captured in human motion videos (Section 3.1.1). These extracted landmarks are then used to construct a dataset for self-supervised feature representation learning (Section 3.1.2). To evaluate the robustness of CLUMM against outliers and its applicability to anomaly or outlier detection, we deliberately introduce outliers into the training dataset (Section 3.2).

### 3.1. CLUMM Training

We begin by extracting image frames from human motion videos to enable human motion recognition on raw camera data without manual data labeling. We then use MPP to identify the spatial locations of 33 body joints, as shown in Figure 2. Of the 33 landmarks detected, 10 landmarks of interest are selected based on careful evaluation by ergonomics experts and evidence from existing work to construct a dataset for motion recognition (Section 3.1.1). We utilize a modified version of SimCLR, a contrastive SSL framework for image representation learning, to learn robust representations from the constructed dataset (Section 3.1.2). SimCLR identifies intrinsic motion types in the dataset by grouping similar representations in the feature space and pushing dissimilar ones apart. Finally, we train a simple multiclass logistic regression model on top of the learned representations in a few-shot learning setting, using transfer learning on a small subset of labeled data to validate the generalizability of the representations on downstream motion recognition tasks (Section 3.1.3).

#### 3.1.1. Pose Landmark Extraction

We employ MPP [19] to estimate poses and track landmark locations from image frames obtained from motion videos. MPP is an open-source framework created by Google to estimate high-fidelity 2D and 3D coordinates of body joints. MPP uses BlazePose [20], a lightweight pose estimation network, to detect and track 33 3D body landmarks from videos or images, as depicted in Figure 2. These landmark positions approximate the location of each identified body part in either image or world coordinates. Research in [40] has shown that using joint coordinates as inputs for DL training leads to better results than using raw RGB images. To avoid learning spurious correlations [59] and optimize the performance of our learned representations, we aim to reduce the influence of environmental factors like lighting, contrast, and irrelevant features that might affect the deep learning network and conceal the features of interest. We achieve this by using MPP to extract pose landmarks pertinent to the specific activities of interest, ultimately revealing the position of various joints in the human body. BlazePose can also handle occlusions, thus making our approach robust to occlusions in front of the detected human. We extract 10 landmarks from the upper and lower limbs (Table 1) relevant to the motion tasks we study in this paper, denoted as an index set I. These landmarks were chosen based on evidence from prior studies that identified and used specific landmarks relevant to the task they investigated [5,44]. This approach helps to focus on the human movement and avoid learning background information. We obtain pose landmarks in normalized image coordinates to make our extraction agnostic to body shape or type.

We extract the normalized X, Y, and Z coordinates of each image frame’s ten landmarks of interest and quantize them into a feature vector of 30 elements. Missing coordinates are replaced by 0 to indicate their absence. We finally construct a dataset, a matrix of each frame’s feature vectors (see Figure 2).

Assuming $L_{i}=\left(x_{i},y_{i},z_{i}\right)$ are the cartesian coordinates of the ith landmark. For each image frame $D_{j},j=\{1,\;2,\;\ldots ,\;m$}, the feature vector $V_{j}$ extracted is $V_{j}=\left[L_{j_{1}},L_{j_{2}},\ldots ,L_{j_{10}}\right]=\left[\left(x_{j_{1}},y_{j_{1}},z_{j_{1}}\right),\ldots ,\left(x_{j_{10}},y_{j_{10}},z_{j_{10}}\right)\right]$. Here, *m* is the number of images. The entire dataset is shown in Equation (1).(1)$$v=\left[\begin{array}{c} V_{1}\\ V_{2}\\ \vdots \\ V_{m} \end{array}\right]=\left[\begin{array}{c} \begin{array}{ccc} \left(x_{1_{1}},y_{1_{1}},z_{1_{1}}\right) & \cdots & \left(x_{1_{10}},y_{1_{10}},z_{1_{10}}\right)\\ \left(x_{2_{1}},y_{2_{1}},z_{2_{1}}\right) & \cdots & \left(x_{2_{10}},y_{2_{10}},z_{2_{10}}\right) \end{array}\\ \begin{array}{ccc} \vdots & \ddots & \vdots \\ \left(x_{m_{1}},y_{m_{1}},z_{m_{1}}\right) & \cdots & \left(x_{m_{10}},y_{m_{10}},z_{m_{10}}\right) \end{array} \end{array}\right]$$

#### 3.1.2. Contrastive Self-Supervised Representation Learning

We use the SIMCLR contrastive learning framework [21] to learn robust representations in the pose data constructed in Section 3.1.1. SimCLR is a straightforward contrastive SSL framework that maximizes and minimizes the agreement between positive and negative pairs [21]. Positive pairs are generated by applying random augmentations on the same sample. SimCLR is made up of four key components:

**Random augmentation module**: This module applies random data augmentations on input samples $V_{i},\;i=\left\{1,\;2,\;\ldots ,\;m\right\}$. It performs two transformations on a single sample, resulting in two correlated views ${\tilde{V}}_{i_{1}}$ and ${\tilde{V}}_{i_{2}}$ treated as a positive pair.**Encoder module** $f\left(\cdot \right)$: This module uses a neural network to extract latent space encodings of the augmented samples ${\tilde{V}}_{i_{1}}$ and ${\tilde{V}}_{i_{2}}$. The encoder is model agnostic, allowing various network designs to be used. The encoder produces output $h_{i_{1}}=f\left({\tilde{V}}_{i_{1}}\right)$ and $h_{i_{2}}=f\left({\tilde{V}}_{i_{2}}\right)$, where f is the encoder network.**Projector head** $g\left(\cdot \right)$: This small neural network maps the encoded representations into a space where a contrastive loss is applied to maximize the agreement between the views [16]. A multi-layer perceptron is used, which produces output $\psi_{i_{1}}=g\left(h_{i_{1}}\right)$ and $\psi_{i_{2}}=g\left(h_{i_{2}}\right)$, where g represents the projector head and $h_{i}$ represents the output of the encoder module.**Contrastive loss function**: This learning objective maximizes the agreement between positive pairs.

SimCLR was initially designed for images. Therefore, we make the following modifications to accommodate our transformed pose landmark data.

A.Data Augmentation

We apply two random transformations on our input data to compose the augmentations for our SimCLR training.

**Random Jitter** $t_{1}\left(\cdot \right)$: We apply a random jitter on the input samples using a noise signal drawn from a normal distribution with a mean of zero and a standard deviation of 0.5, i.e., $\varepsilon_{i}\sim N\left(0,0.5^{2}\right)$. Thus, for each input sample $V_{i},$ we obtain ${\tilde{V}}_{i_{1}}=t_{1}\left(V_{i}\right)=V_{i}+\varepsilon_{i},\;i=\left\{1,\;2,\;\ldots ,\;m\right\}$.**Random scaling** $t_{2}\left(\cdot \right)$: We scale samples with a random factor drawn from a normal distribution with a mean zero and a standard deviation of 0.2, i.e., $\eta_{i}\sim N\left(0,0.2^{2}\right)$. Therefore, for each input sample $V_{i}$, we obtain ${\tilde{V}}_{i_{2}}=t_{2}\left(V_{i}\right)=V_{i}\cdot \eta_{i},\;i=\left\{1,\;2,\;\ldots ,\;m\right\}$.

These transformations, shown in the literature to improve the feature representation performance of contrastive learning models [60,61,62], are composed to form the augmentation module of our modified SimCLR.

B.Encoder Module

We use a pre-trained ResNet model as our encoder to obtain latent space representations of the input samples. To accommodate our data, which consists of a single channel instead of the three channels expected for RGB images, we modify the first convolutional layer of the ResNet. Specifically, we adjust the input shape and update the weights of the first convolutional layer to align with the dimensions of our single-channel input from MediaPipe. The ResNet model was introduced by He et al. [48] to address the vanishing gradient problem associated with deep networks by introducing the residual or skip connection, which directly adds the input ξ to the outputs $F\left(\xi ,W\right)$ of a network. Thus, the residual block is represented as $\phi =F\left(\xi ,W\right)+\xi$, where W is the weight matrix of a layer. We leverage this in our encoder to learn robust representations without degradations.

C.Projector Head

We use a simple multilayer perceptron (MLP) to transform the learned representations from the encoder module into a space where we apply a contrastive loss to maximize the agreement between positive pairs. Our projection head is a three-layer MLP consisting of a linear layer followed by a ReLU activation function and a final linear layer.

D.Contrastive Loss Function

CLUMM uses the Normalized Temperature-scaled cross entropy (NT-Xent) loss [63] from SimCLR. Given two different augmentations ${\tilde{V}}_{i_{1}}$ and ${\tilde{V}}_{i_{2}}$ of an input sample, NT-Xent aims to bring these views closer together in the feature space while pushing views from different samples apart. Assuming $\psi_{i_{1}}$ and $\psi_{i_{2}}$ are the embeddings of ${\tilde{V}}_{i_{1}}$ and ${\tilde{V}}_{i_{2}}$ respectfully, after passing through the projector head, NT-Xent is defined as:(2)$$L_{i_{1},i_{2}}=-\log\frac{e^{\frac{CoSim\left(\psi_{i_{1}},\psi_{i_{2}}\right)}{\tau }}}{\sum_{k=1}^{2b}1_{\left[k\neq i_{1}\right]}e^{\frac{CoSim\left(\psi_{i_{1}},\psi_{i_{k}}\right)}{\tau }}}$$
where

CoSim is the cosine similarity between the two embeddings.

b is the batch size.

τ is the temperature parameter that controls the sharpness of the distribution.

1 is the indicator function that evaluates to 1 when $k\;\neq i_{1}$.

Algorithm 1 summarizes the representation learning process.
**Algorithm 1**: CLUMM feature representation learningInput: Human Motion Videos D1,D2,…,Dn, landmark extractor M (MediaPipe in this context), landmark indices I, constant τ, batch size b, structure of f,g, and augmentation functions t1,t2# Step 1: pose estimation and dataset constructionfor each video Dj,j=1,2,…,m:      # Extract image frames from video Dj=Dj1,Dj2,…,Djm    for each frame Dji∈vj            # Extract landmarks indexed by I            Lji=MDji=Lji1,Lji2,…,Lji10            # Concatenate the extracted landmarks into a feature vector Vi    **end**    # Construct a feature matrix from feature vectors Vji,i=1,2,…,m    vj=Vj1,Vj2,…,VjmT**end**#Step 2: SimCLR trainingfor sampled minibatch in Vkk=1b∈vj,j=1,2,…,n    for each sample k∈1,…,b          # Apply the two augmentation functions to each sample            V~k1=t1Vk,V~k2=t2Vk            # Pass augmented samples through encoder f            hk1=fV~k1,hk2=fV~k2            # Pass encoded representations through projection head g            ψk1=ghk1,ψk2=ghk2    **end**    for each pair k1,k2∈1,…2b         # Compute pairwise similarity            Sk1,k2=CoSimψk1,ψk2         # Compute NT-Xent loss in Equation (2)    **end**# Update networks f and g to minimize the loss**end**return encoder f and throw away g

#### 3.1.3. Finetuning with SoftMax Logistic Regression

We assess the generalization performance of our learned representations by fine-tuning a multinomial logistic regression model [64] in a transfer learning setting. Multinomial logistic regression extends the standard logistic regression [65] for classification problems with more than two classes. Given a dataset Vi,yi i=1n where $V_{i}\in R^{d}$ is the ith input feature vector and $y_{i}\in \;\left\{1,2\ldots ,K\right\}$ is the corresponding class label with K representing the number of classes, multinomial logistic regression computes the probability of a sample $V_{i}$ belonging to class k using the SoftMax [66] represented as Equation (3)(3)$$P\left(y_{i}=k|V_{i}\right)=\frac{e^{\left(\phi_{k}\left(V_{i}\right)\right)}}{\sum_{j=1}^{K}e^{\left(\phi_{j}\left(V_{i}\right)\right)}}$$
where $\phi_{k}$ is the raw calculated score for each class given by(4)$$\phi_{k}\left(V_{i}\right)=w_{k}^{T}V_{i}+b_{k}$$
where b is the bias term, $w_{k}$ is a weight vector associated with a class k.

Given a single training instance $\left(V_{i},y_{i}\right)$, the loss function is represented as(5)$$l\left(W,b\right)=-logP\left(\left(y_{i}\right|{\;V}_{i}\right)$$
with $W=\left[w_{1}^{T},w_{2}^{T},\ldots ,w_{K}^{T}\right]$, which generalizes across all training samples as(6)$$J\left(W,b\right)=-\frac{1}{n}\sum_{i=1}^{n}logP\left(\left(y_{i}\right|{\;V}_{i}\right)$$The parameters W and b are learned by minimizing the loss function using an optimization technique such as gradient descent or its variants [67].

We manually categorize a subset of our dataset into three activity classes, *idle*, *lift*, and *bend*, and randomly split the dataset into a training and a testing set (distinct from those used for pretext training). We fix the encoder weights of CLUMM for transfer learning and train only the linear layer connected to the encoder. We conduct 5-fold cross-validation to prevent selection bias and ensure a dependable evaluation of the performance of our logistic model.

### 3.2. Impact of Outliers on CLUMM Performance

Using ML/DL methods for human motion recognition requires training data. For the proposed CLUMM, training data containing several fixed motion types (classes) are used to train SimCLR to obtain the feature embeddings. Introducing unexpected motion types, in addition to the fixed ones, would result in “outliers” in the training data, potentially impacting SimCLR’s feature extraction performance and, thus, the classification accuracy. We investigate the effects of these outliers on our proposed model by first clustering the features of interest and calculating the distance from outliers to inliers. We leverage the K-Means clustering algorithm to cluster the extracted coordinates (features). The K-Means algorithm minimizes the within-cluster sum of squared distances (WCSS), defined as:(7)$$WCSS=\sum_{i=1}^{c}\sum_{V\in C_{i}}|V-\mu_{i}{|}^{2}$$
where

c is the number of clusters,

$C_{i}$ is the set of points belonging to cluster i,

$\mu_{i}$ is the centroid of cluster i,

$V=\left[V_{1},V_{2},\ldots ,V_{30}\right]$ is a data point (feature vector of length 30 here),

$|V-\mu_{i}{|}^{2}$ is the squared Euclidean distance between V and $\mu_{i}$.

In our case, the coordinates are clustered into c=3 clusters and the centroids are updated to minimize WCSS. After performing K-Means clustering, the Euclidean distance between each point V and the closest centroid $\mu_{i}$ is computed using:(8)$$d\left(V,\mu_{i}\right)=\sqrt{{\left(V_{1}-\mu_{i1}\right)}^{2}+{\left(V_{2}-\mu_{i2}\right)}^{2}+\cdots +{\left(V_{30}-\mu_{i30}\right)}^{2}}$$
where $V_{1},V_{2},\ldots ,V_{30}$ are the coordinates of the point V (where we use V to represent the $V=\left[\left(x_{1},y_{1},z_{1}\right),\ldots ,\left(x_{10},y_{10},z_{10}\right)\right]$ defined in Equation (1)), $\mu_{i1},\mu_{i2},{\ldots ,\mu }_{i30}$ are the coordinates of the centroid $\mu_{i}$.

Outliers are data points whose distance from the nearest centroid exceeds a threshold. The threshold is calculated as:(9)$$t=\mu_{d}+2\sigma_{d}$$
where $\mu_{d}$ is the mean of the distances from points to their nearest centroid (see Equation (10)), and $\sigma_{d}$ is the standard deviation of those distances.

Points with $d\left(V,\mu_{i}\right)>t$ are considered outliers. For each cluster, the following metrics are computed:

The average distance from all points in a cluster i to their respective centroid

(10)$$\mu_{d}^{\left(i\right)}=\frac{1}{\left|C_{i}\right|}\sum_{V_{j}\in C_{i}}d\left(V_{j},\mu_{i}\right)$$
where

$\mu_{i}$ is the centroid of the cluster, which is the mean of all points in the cluster in terms of their coordinates.

$\mu_{d}^{\left(i\right)}$ is the average distance between each point in the cluster and the centroid $\mu_{i}$. It quantifies how tightly clustered the points are around the centroid.

$d\left(\cdot ,\cdot \right)$ is the distance between the data point and the centroid, typically computed using the Euclidean distance formula.

$\left|\cdot \right|$ is the cardinality of a set measuring the size (or the number of data points) of the set [68].

2The maximum distance of any point in a cluster i to the centroid:(11)$${Μ}_{d}^{\left(i\right)}=\underset{V_{j}\in C_{i}}{\max}\left(d\left(V_{j},\mu_{i}\right)\right)$$

$M_{d}$ indicates how spread out or far away the most distant point is from the cluster’s center.

Since our framework relies on contrastive learning, where the performance of the loss is dependent on the selection of positive and negative pairs [69], the presence of outliers in the training data is likely to distort the process by presenting false positive pairs or hard negatives, causing the model to emphasize irrelevant features which will affect the quality of the learned representations and the generalization of the model to downstream tasks. Given the contrastive loss in Equation (2), the presence of outliers introduces noisy or irrelevant negative pairs, increasing the denominator and making it harder to distinguish true positive pairs. Additionally, false positives introduced by outliers distort the similarity metric $CoSim\left(\psi_{i_{1}},\psi_{i_{2}}\right)$ by lowering the similarity and subsequently increasing the loss. Assuming a binary indicator $\omega_{i}\in 0,1$ where $\omega_{i}=1$ indicates the presence of an outlier and $\omega_{i}=0$ as a regular sample, we can rewrite the contrastive loss Equation (2) with outliers as:(12)$$L_{i_{1},i_{2}}=-\log\frac{e^{\frac{CoSim\left(\psi_{i_{1}},\psi_{i_{2}}\right)}{\tau }}}{\sum_{k=1}^{2b}1_{\left[k\neq i_{1}\right]}e^{\frac{CoSim\left(\psi_{i_{1}},\psi_{i_{k}}\right)}{\tau }}+\lambda \delta \omega_{i}}$$
where

λ is the mean distance of an outlier to the cluster centroid.

δ is a weight controlling the impact of the outlier on the denominator.

This modification of the contrastive loss shows that outliers $\omega_{i}=1$ increase the denominator and consequently the overall loss, affecting the ability of the network to learn good representations.

### 3.3. Summary of CLUMM

As shown in Figure 3, our proposed CLUMM framework first extracts frames from videos. Then, it performs a pose extraction step (Section 3.1.1) to produce quantized input samples. These are sent into a contrastive learning step (Section 3.3) to learn feature representations directly from the unlabeled input. To demonstrate the robustness of the proposed framework, we finetune a simple multinomial logistic regression model on the learned representation using a small set of manually labeled data. The result of this experiment is shown in Section 4.

### 3.4. Note to Practitioners

In this paper, we approach the human motion recognition problem from a self-supervised approach using skeletal coordinates of various motion types across various application domains. Findings from our experiments show that our approach can effectively identify intrinsic motion types in the data, which is a benefit of self-supervised learning. Our framework can detect different motion types on the fly as they are introduced to the training data. This ability to capture intrinsic motions from the data makes our approach applicable to motion recognition, outlier analysis, and anomaly detection by fine-tuning on labeled data with transfer learning.

## 4. Case Study

To evaluate CLUMM’s effectiveness in motion recognition, we conducted experiments on a custom dataset simulating various tasks in a workplace setting. The dataset comprises videos captured in a laboratory environment using an optical camera. The experimental design involved human subjects performing tasks with boxes of varying sizes. These tasks included lifting a box, inserting a box into another box, and placing a box on a surface. The actions were performed either randomly or systematically using a standardized numbering system. Video recordings of two participants (age 27 ± 1.41 years, arm length 0.70 ± 0.014 m, height 1.74 ± 0.014 m) were recorded at 30 frames per second with standardized resolution and dimensions. The frame rate was determined by the naturalistic movements captured during the recording and was not explicitly standardized between participants. This approach allowed us to analyze the full range of behaviors as they occurred naturally, without imposing artificial constraints on the data. While we did not control the frame rate or number of frames directly, this method ensures that the analysis aligns closely with the spontaneous and realistic context of the studied interactions. To enhance the dataset and eliminate blind spots, videos were recorded from three angles (left, center, and right), providing multiple perspectives. However, only videos from the center angle were used in our experimental procedure. Figure 4 shows the camera placement with respect to the participants in the experimental setup. The recorded videos were time-synchronized and denoised to ensure frame consistency across the dataset.

The experimental procedure was approved by the Institutional Review Board of Arizona State University. Each participant reviewed and signed an IRB-approved informed consent before participation.

### 4.1. Data Preparation

We extracted image frames from three different videos of a human subject performing multiple tasks. These frames contained distinct actions repeated across each task. The unlabeled image frames were shuffled and preprocessed for CLUMM training, followed by the pose extraction pipeline described in Section 3.1.1. This process eliminated background effects, such as the presence of boxes and variations in lighting, as well as differences in the size and height of the human subject. These steps ensured that the representations learned from the data were generic, invariant to scale and lighting conditions, and generalized across diverse operational environments. To ensure consistency, the views were kept the same as the views used in unsupervised training. Enough frames are crucial to effectively capturing the temporal dynamics of the motion types, which directly affects the robustness of the representations learned in the next stage of the pipeline. However, increasing the number of frames increases the computational and memory demands which may pose practical limitations depending on the resources available.

### 4.2. CLUMM Validation on Custom Data

We trained the modified SimCLR (Section 3.1.2) to learn representations directly from the unlabeled pose dataset. For the encoder, we experimented with two pre-trained models, ResNet18 and ResNet50, and employed a three-layer MLP with an output dimension of 128 as the projector head (Section 3). The AdamW optimizer was used with an initial learning rate of 0.005 and a cosine learning rate decay schedule for pretraining over 500 epochs. Due to limited data availability, we used a small batch size of 64. This training step was designed to learn robust representations that could serve as initial weights for downstream motion classification. Leveraging SimCLR, we grouped features representing the same motion types while features of different motion types were pushed apart. This process effectively captured the intrinsic variability of motion types in the dataset.

To explicitly categorize the motion types, we manually labeled a subset of the dataset into three activity classes: *idle*, *lift*, and *bend*. The labeled dataset was then randomly split into a training set of 1500 frames and a test set of 916 frames (distinct from the frames used for pretext training). Since the learned representations from the modified SimCLR are feature embeddings rather than class labels, we trained a simple multinomial logistic regression model on the labeled dataset to identify motion categories and assess the efficiency of the learned representations. This was performed in a transfer learning setting where the encoder weights of the SimCLR component were frozen, and only the logistic regression layer was trained. Thus, SimCLR functioned as a feature extractor for the logistic regression model. The logistic regression model was trained with 5-fold cross-validation for 100 epochs using the Adam optimizer with a learning rate of 0.01 and a multistep learning rate decay. We compare the motion recognition performance of CLUMM using ResNet18 and ResNet50 backbones to the performance of baseline ResNet18 and ResNet50 transfer learners on the same dataset; the initial layers of the ResNets are frozen, with only the last fully connected layer being trained. We keep everything else the same as the fine-tuning process with CLUMM. Table 2 shows the performance of our fine-tuned CLUMM and baseline ResNet models (the best performances have been bolded). CLUMM outperformed the baseline models with either ResNet18 or ResNet50 backbones on our custom dataset, indicating the enhanced feature extraction by CLUMM due to its self-supervised learning capability.

### 4.3. CLUMM Validation on Public Data

To validate the generalizability of our proposed methodology, we conducted an evaluation using the publicly available UCF Sports Action Dataset https://www.crcv.ucf.edu/data/UCF_Sports_Action.php (accessed on 2 January 2025), a widely recognized benchmark for motion recognition. This dataset contains videos of ten distinct actions: *diving, golf swinging, kicking, weightlifting, running, skateboarding, swinging (on a rope), horse riding, walking, and high jumping*. These activities are captured in realistic environments, exhibiting significant variations in camera angles, participants, lighting conditions, sizes, and backgrounds. Figure 5 illustrates images from the same class displaying different characteristics. This diversity makes the dataset both challenging and representative of real-world scenarios. We focused our experiments on four motion types: *running, skating, lifting, and walking*. These motions feature centered perspectives and closely align with the action classes in our laboratory-collected dataset. Briefly, 3202 RGB images of varying sizes and resolutions were initially extracted across the four classes. After refinement, which involved removing frames without humans, 3166 image frames were obtained across these four actions. We followed the same methodological procedure outlined in Section 3 and kept all hyperparameters consistent with those used for our custom dataset in Section 4.2. Pose extraction was performed using the pipeline described in Section 3.1.1 to identify key pose landmarks. Figure 6 illustrates the identified landmarks on a sample image. The extracted pose coordinates were then used to fine-tune our modified SimCLR representation learner, with all hyperparameters kept the same as in our first case study to ensure consistency. To explicitly classify the motion types, 30% of the extracted frames were labeled and used to train a multinomial logistic regression model to classify the four motion types. This training was conducted in a transfer learning setting, where the encoder weights of the SimCLR component were frozen, allowing only the logistic regression layer to be trained. In this setup, SimCLR served as a feature extractor for the logistic regression model. The logistic regression model was trained using 5-fold cross-validation for 100 epochs, employing the Adam optimizer with a learning rate of 0.01 and a multistep learning rate decay. Table 2 compares the fine-tuning performance of our model with a baseline ResNet model and our training results from the custom dataset.

### 4.4. Results from Outlier Analysis

We manually introduced outliers into the training data to investigate the effects of outliers on our proposed model. We introduced different motion types from the three motion categories used in this study. We examined 100, 200, and 500 outliers and reported their impact on the performance of CLUMM with ResNet18 backbone (See Table 3). Since outliers may have specific landmark positions similar to the regular motion, we performed outlier analysis for each landmark using the methodology described in Section 3.3. Keeping all our parameters and hyperparameters constant, we performed CLUMM training on the new training datasets with outliers and fine-tuned testing data. As hypothesized in Section 3.3, we observed a decrease in the generalization capacity of CLUMM on the downstream task with the introduction of outliers in the training data. However, we observed that increasing the number of outliers from 200 to 500 did not affect performance by a significant margin. This proves the robustness of the framework to outliers. We show the results of our analysis in Table 3. Figure 4 shows a 2D representation of the embedding space, where the axes are the first and second principal components, demonstrating the grouping of the various motion types intrinsic to the data. The majority of the data instances have been well separated from other motion types. Table 3 Results from outlier analysis using a ResNet18 backbone.

## 5. Conclusions and Future Work

This paper presents a comprehensive framework for human motion analysis, bridging gaps in the implementation of efficient and unobtrusive human motion recognition to promote better human well-being. Our proposed solution leverages MediaPipe and a contrastive learning framework for feature extraction, addressing data-level challenges by streamlining the data labeling and feature extraction processes. Results from fine-tuning domain-specific data demonstrate the effectiveness of our approach in achieving accurate and scalable motion analysis, which can be adapted to other motion-related tasks in complex environments. Our fine-tuned model outperforms a baseline supervised model, showcasing the potential of self-supervised learning to significantly reduce manual labeling efforts.

Although this study achieved impressive results, there are limitations that we plan to address in future work.

Firstly, our proposed framework, CLUMM, performs reasonably well in the UCF Sports Action Dataset. However, it does not achieve accuracy levels comparable to specialized action recognition models such as Inflated 3D ConvNet (I3D) [70], which have reported accuracies exceeding 90% on datasets like UCF101. This limitation arises from CLUMM’s focus on label efficiency and its primary application in unobtrusive motion monitoring scenarios. CLUMM is still in the exploratory stage. Future work will focus on further refinement and fine-tuning on a larger dataset to enhance its performance.

Secondly, the custom dataset was collected in a controlled laboratory environment, which may not fully reflect the dynamics of real-world operations. In our future research, we aim to collect data in natural environments to validate the scalability of our proposed methodology.

Additionally, we plan to extend this work to include multi-camera human sensing and adaptive multi-sensory data fusion. Our future efforts will specifically focus on integrating multiple camera perspectives to capture more comprehensive data and applying fusion technologies to learn simultaneously from various angles. We also intend to investigate how incorporating new data streams can enhance the performance of our representation learning module and enable continuous learning of new motion types.

## Figures and Tables

**Figure 1 sensors-25-01048-f001:**
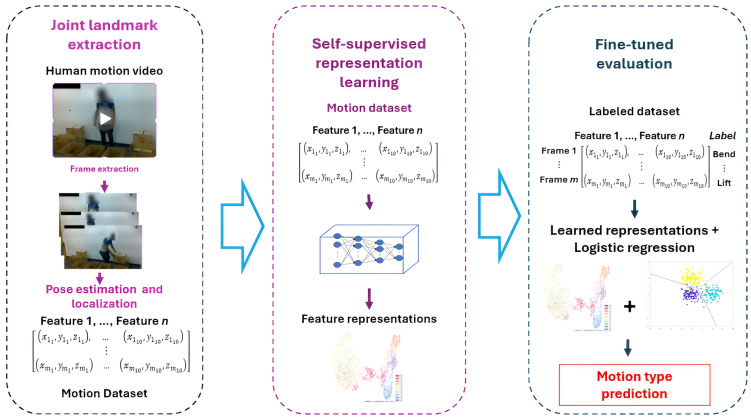
Architecture pipeline for human motion analysis. “Features” are the locations of the extracted landmarks.

**Figure 2 sensors-25-01048-f002:**
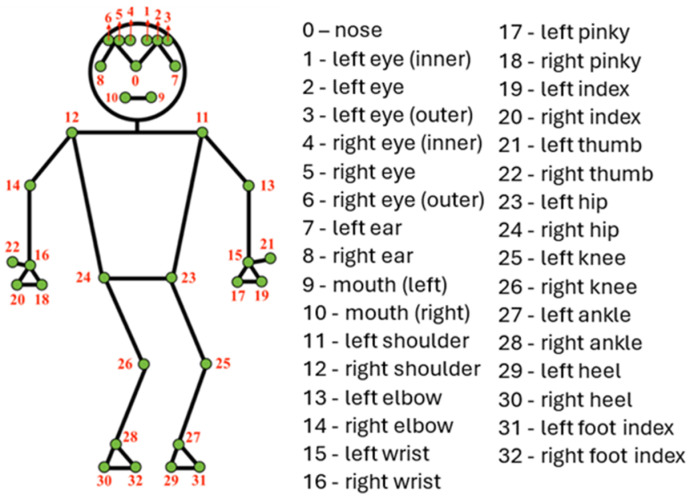
Human Body Pose Landmarks, adapted from MediaPipe [19].

**Figure 3 sensors-25-01048-f003:**
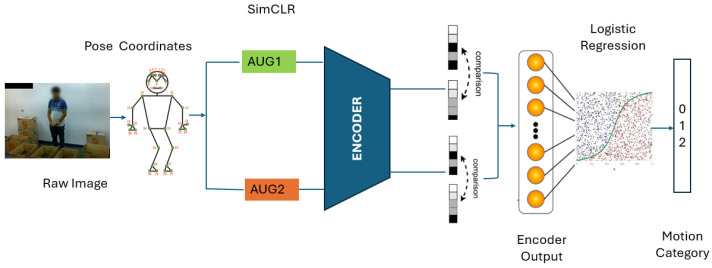
CLUMM Framework.

**Figure 4 sensors-25-01048-f004:**
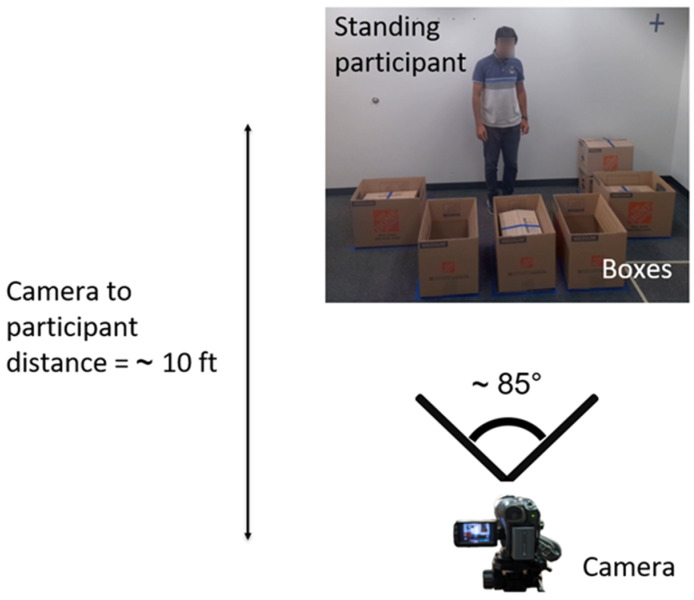
Placement of camera in experimental setup.

**Figure 5 sensors-25-01048-f005:**
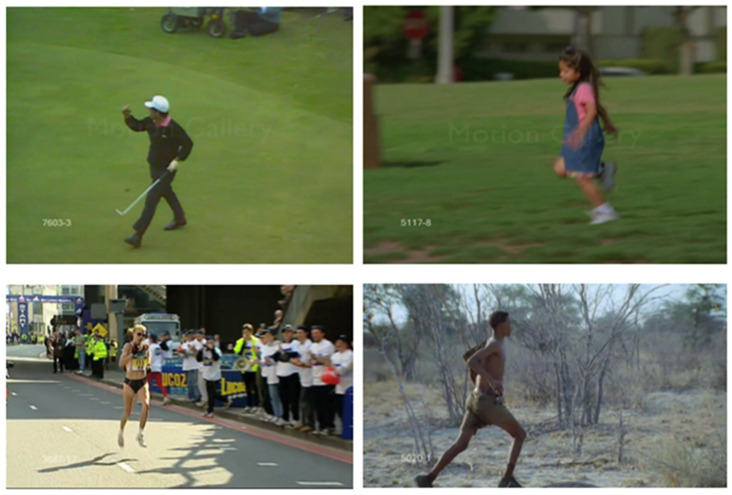
Images showing different motion characteristics from the same class in the UCF Sports Action Dataset.

**Figure 6 sensors-25-01048-f006:**
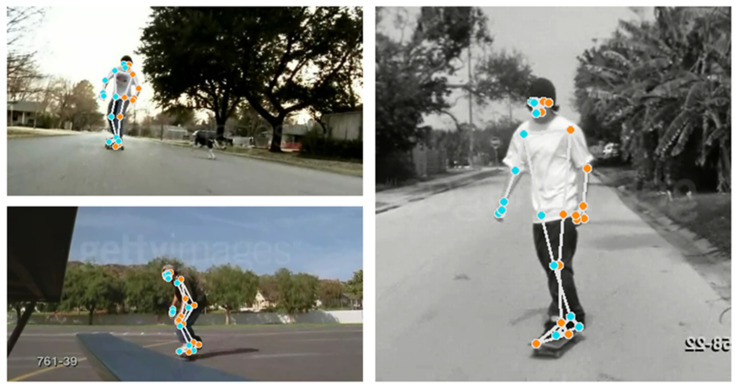
Pose Detection from sample image class in the UCF Sports Action Dataset. Colored dots represent identified pose landmarks.

**Table 1 sensors-25-01048-t001:** Pose Landmarks of Interest, i.e., elements in I.

Pose Number	Representation
11	Left Shoulder
12	Right Shoulder
13	Left Elbow
14	Right Elbow
15	Left Wrist
16	Right Wrist
23	Left Hip
24	Right Hip
25	Left Knee
26	Right Knee

**Table 2 sensors-25-01048-t002:** CLUMM performance comparison with baseline ResNets.

Dataset	Network	Accuracy	Precision	Recall	F1 Score
	ResNet18 Baseline	79.6%	0.801	0.797	0.796
	ResNet50 Baseline	77.3%	0.782	0.773	0.771
Custom	**CLUMM (ResNet50 Backbone)**	**83.5%**	**0.833**	**0.835**	**0.831**
	**CLUMM (ResNet18 Backbone)**	**90.0%**	**0.899**	**0.90**	**0.899**
UCF Sports Action	ResNet18 Baseline	78.9%	0.839	0.826	0.829
	ResNet50 Baseline	77.3%	0.818	0.777	0.791
	**CLUMM (ResNet50 Backbone)**	**86.9%**	**0.869**	**0.865**	**0.865**
	**CLUMM (ResNet18 Backbone)**	**88.2%**	**0.882**	**0.879**	**0.878**

Bold texts represent our customized model.

**Table 3 sensors-25-01048-t003:** Results from outlier analysis using a ResNet18 backbone.

Number of Outlier Images	Percentage of Outlier Landmarks	Mean Outlier Distance	Max Outlier Distance	Accuracy	Precision	Recall	F1-Score
None	-			90.00%	0.899	0.90	0.899
100	3.5%	0.30	0.80	86.76%	0.867	0.867	0.865
200	3.1%	0.30	0.81	84.48%	0.845	0.844	0.843
500	3%	0.31	0.83	84.64	0.845	0.846	0.846

## Data Availability

The human subject experimental data are not publicly available due to privacy reasons. The UCF Sports Action data are available at https://www.crcv.ucf.edu/data/UCF_Sports_Action.php accessed on 22 November 2024. Our code and case study for the UCF Sports Action data are available on GitHub at https://github.com/linhobs/Clumm accessed on 22 November 2024.

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
