# Peer review of "CLUMM: Contrastive Learning for Unobtrusive Motion Monitoring"

_sensors, 2025, doi:10.3390/s25041048_

Round 1
Reviewer 1 Report
Comments and Suggestions for Authors
The reviewer believes that the following issues are the fatal problems of this article:
The innovation level of this article is too low. Although section 3.1 is quite lengthy, it can be summarized in just one sentence: firstly, the author uses Mediapipe to identify the spatial positions of 33 joints, and secondly, the author uses the contrastive learning framework SimCLR to recognize actions. So, the reviewer would like to ask, is the SimCLR framework just right for the current motion detection task? Isn't it necessary to make some changes based on the current task? Of course, the changes mentioned by the reviewer are not just modifications to the 460 line network input.
In section 3.1, the author provides a lengthy and detailed introduction to Mediapipe and contrastive learning. So the reviewer wants to ask, shouldn't the third part of the paper include original content written by the author? Why do we need to provide a lengthy introduction to our existing work? Is it to take up space?
The method proposed by the author is a simple patchwork of existing methods. Algorithm 1 can support the reviewer's viewpoint. An original method, or an original neural network, must include all the parts of extracting input image features, recognizing joint positions, and classifying actions. One end is the input image, and the other end is the action recognition result, and Algorithm 1 is divided into two parts. This is clearly a combination of two existing methods.
The experimental results in this article are not convincing. The author did not compare the proposed method with other studies. The other research mentioned by the reviewer refers to the methods of human pose recognition proposed in other papers, rather than comparing the framework proposed by oneself with other frameworks proposed by oneself. It is very necessary to compare one's own research with others' research. If the accuracy of others is already higher than one's own research, then one's own research is meaningless.
In addition, there are some details in this article
The literature review is too long. It is suggested to discuss the literature directly related to this study. Assuming that the author's research belongs to a mixed method of deep learning and traditional features, this section only needs to discuss this method. The research on wearable sensors that has been criticized in this study should be briefly mentioned in the introduction section. There is no need to restate in the literature review section.
At the beginning of the literature review section, the author should refine and classify the literature directly related to this study, and classify similar research methods or techniques into one category, so that readers can quickly understand the overview of the entire research task.
Suggestion 2.2: Self supervised learning of technology related popular science should not be included in the main text, as readers will learn on their own. Literature review is to let the author understand the general picture of the research field and prove that his research is progressiveness, rather than let the reader learn knowledge.
Reviewer 2 Report
Comments and Suggestions for Authors
This paper proposes a framework that enhances human monitoring and motion recognition in manufacturing environments by utilizing unobtrusive camera-based sensing. By employing a contrastive learning approach, using SIMCLR algorithm, they aim to extract valuable information from raw images without the need for extensive manual labeling. The idea is simple and direct, it naturally appears to be a valuable application of machine learning, also the use of k-means is atractive. However, I have some observations about the original contribution of this work.
I have some comments:
1. Page 3. Explain if the number of frames affect procedure in Figure 1.
2. P.8. It lacks a study of self-supervised models applied to action recognition with cameras. For example:
Lin, W., Ding, X., Huang, Y., & Zeng, H. (2023). Self-supervised video-based action recognition with disturbances. IEEE Transactions on Image Processing, 32, 2493-2507.
Dave, I. R., Chen, C., & Shah, M. (2022). Spact: Self-supervised privacy preservation for action recognition. In Proceedings of the IEEE/CVF Conference on Computer Vision and Pattern Recognition (pp. 20164-20173).
Lin, L., Song, S., Yang, W., & Liu, J. (2020, October). Ms2l: Multi-task self-supervised learning for skeleton based action recognition. In Proceedings of the 28th ACM international conference on multimedia (pp. 2490-2498).
Seo, M., Cho, D., Lee, S., Park, J., Kim, D., Lee, J., ... & Choi, D. G. (2021). A self-supervised sampler for efficient action recognition: Real-world applications in surveillance systems. IEEE Robotics and Automation Letters, 7(2), 1752-1759.
Yang, Y., Liu, G., & Gao, X. (2022). Motion guided attention learning for self-supervised 3D human action recognition. IEEE Transactions on Circuits and Systems for Video Technology, 32(12), 8623-8634.
Ghelmani, A., & Hammad, A. (2023). Self-supervised contrastive video representation learning for construction equipment activity recognition on limited dataset. Automation in Construction, 154, 105001.
Dedhia, U., Bhoir, P., Ranka, P., & Kanani, P. (2023, September). Pose Estimation and Virtual Gym Assistant Using MediaPipe and Machine Learning. In 2023 International Conference on Network, Multimedia and Information Technology (NMITCON) (pp. 1-7). IEEE.
Etc.. there are multiple near works.
Naturally, you should compare your proposal against the other works.
3. P9. What is m?
4. P10. "𝑖 = 1, 2, … ,m" -> "𝑖 = {1, 2, … ,m}". The same for similar definitions.
5. P10. "We modify the first convolutional layer of the ResNet to ac- 459
commodate our data"-> What where the modification?
6. P11. Equation 2 (and 12!) is wrong. Check the subindexes.
7. P11. How do you consider the number of frames for video?. Are the same for each one?
8. P14. How many participants?
9. P14. How are recorded the video in relation to angles?. Perhaps you should show a diagram with the locations of cameras.
10. P15. Not necessary "Notably, no labels were assigned 614
to the dataset at this stage.".
11. P15. pre-text -> pretext
12. P15. The views of supervised data are the same of unlabelled data?. Indicate the differences.
13. P15. The experiments are not conclusive. The dataset is very limited. You should apply over a public dataset, at least one, but it could better to use two datasets. This dataset should be about action recognition. The idea is to eliminate possible bias given that authors are the owners of datasets.
14. P16. Publish datasets and demo of codes. All machine learning research should be replicable.
Round 2
Reviewer 1 Report
Comments and Suggestions for Authors
The reviewer believes that this paper can be published.
Author Response
The authors thank the reviewer for recognizing the value of this work. Sincerely appreciate it.
Reviewer 2 Report
Comments and Suggestions for Authors
This paper presents a self-supervised learning model for action recognition. The authors are thanked for addressing all required points. The contribution is of a minor to medium level given the use of well-known methods. However, the video analysis community may find this proposal of interest as a reference. On the other hand, there are areas for improvement.
Some comments:
1. Figure 1 has low resolution.
2. What is the 'feature' in Figure 1? Is it the position? Please clarify.
3. The performance on UFC is significantly below specialized action recognition models; for example, on UFC100 (1000 actions), accuracies above 90% have been reported in a somewhat outdated paper..
https://openaccess.thecvf.com/content_cvpr_2017/papers/Carreira_Quo_Vadis_Action_CVPR_2017_paper.pdf
The authors should alert the reader that the proposed model is not competitive.
4. The baseline ResNet-50 seems to perform worse than ResNet-18. This is counterintuitive. Why?.
5. It is mandatory to publish datasets and code demos, including baselines given the strange results. All machine learning research should be reproducible.
Round 3
Reviewer 2 Report
Comments and Suggestions for Authors
I appreciate the work of authors and effort in replicating the questions. All points have been solved. In particular, making the code available to the community for proper evaluation of this work is highly valued in my opinion.